# Green Copolymers and Nanocomposites from Myrcene and Limonene Using Algerian Nano-Clay as Nano-Reinforcing Filler

**DOI:** 10.3390/polym14235271

**Published:** 2022-12-02

**Authors:** Hodhaifa Derdar, Geoffrey Robert Mitchell, Artur Mateus, Sarra Chaibedraa, Zinelabidine Otmane Elabed, Vidhura Subash Mahendra, Zakaria Cherifi, Khaldoun Bachari, Redouane Chebout, Rachid Meghabar, Amine Harrane, Mohammed Belbachir

**Affiliations:** 1Centre de Recherche Scientifique et Technique en Analyses Physico-Chimiques (CRAPC), BP 10 384, Siège ex-Pasna Zone Industrielle, Bou-Ismail CP 42004, Algeria; 2Laboratoire de Chimie des Polymères (LCP), Département de Chimie, FSEA, Oran1 University, Ahmed Benbella BP N° 1524 El M’Naouar, Oran 31000, Algeria; 3Centre for Rapid and Sustainable Product Development, Institute Polytechnic of Leiria, 2430-028 Marinha Grande, Portugal; 4School of Chemistry Food Science and Pharmacy, University of Reading, P.O. Box 217, Reading RG6 6AD, UK; 5Department of Chemistry, FSEI University of Abdelhamid Ibn Badis, Mostaganem 27000, Algeria

**Keywords:** myrcene, limonene, copolymerization, Maghnite, nanocomposites

## Abstract

In this work, we report a new facile method for the preparation of myrcene-limonene copolymers and nanocomposites using a Lewis acid as a catalyst (AlCl_3_) and organo-modified clay as a nano-reinforcing filler. The copolymer (myr-co-lim) was prepared by cationic copolymerization using AlCl_3_ as a catalyst. The structure of the obtained copolymer is studied and confirmed by Fourier Transform Infrared spectroscopy, Nuclear Magnetic Resonance spectroscopy, and Differential Scanning Calorimetry. By improving the dispersion of the matrix polymer in sheets of the organoclay, Maghnite-CTA^+^ (Mag-CTA^+^), an Algerian natural organophilic clay, was used to preparenanocomposites of linear copolymer (myr-co-lim). In order to identify and assess their structural, morphological, and thermal properties, the effect of the organoclay, used in varyingamounts (1, 4, 7, and 10% by weight), and the preparation process were investigated. The Mag-CTA^+^ is an organophylic montmorillonite silicate clay prepared through a direct exchange process in which they were used as green nano-reinforcing filler. The X-ray diffraction of the resulting nanocomposites revealed a considerable alteration in the interlayer spacing of Mag-CTA^+^. As a result, interlayer expansion and myr-co-lim exfoliation between layers of Mag-CTA^+^ were observed. Thermogravimetric analysis provided information on the synthesized nanocomposites’ thermal properties. Fourier transform infrared spectroscopy and scanning electronic microscopy, respectively, were used to determine the structure and morphology of the produced nanocomposites (myr-co-lim/Mag). The intercalation of myr-co-lim in the Mag-CTA^+^ sheets has been supported by the results, and the optimum amount of organoclay needed to create a nanocomposite with high thermal stability is 10% by weight. Finally, a new method for the preparation of copolymer and nanocomposites from myrcene and limonene in a short reaction time was developed.

## 1. Introduction

Creating polymers from renewable monomers has been the focus of numerous ongoing research projects worldwide. The majority of research has been conductedon polymers made from renewable resources among the various types of polymers that have been investigated [1,2,3]. The most effective investigations into terpene reactions in organic chemistry synthesis have been published [4], although there are currently few instances of them being used in polymer science. A monocyclic terpene called limonene is found in many essential oils made from citrus rinds and is used as a green solvent as well as an additive to meals, drinks, and cosmetics for flavor and aroma. Limonene contains double bonds, which give the bifunctionality required for polymerization, and because it is an allylic monomer (CH_2_ = CH-CH_2_Y), limonene is of great importance in the field of polymerization [5,6].The results of a literature search show that attempts have been made by chemists to create a substitute for polyterpenes derived from petroleum distillates [7]. However, no such substitute has yet been created because most terpenes are not homopolymerized in the absence of a solvent [8], and have low stabilization energies between monomers and transition state radicals [9], with the exception of β--pinene and limonene, which were polymerized by clay [10], numerous items, including cosmetics, food additives, medicines, and even green solvents, use limonene extensively [11].

Numerous essential oils contain myrcene, which is more properly categorized as monoterpene. Myrcene is a key ingredient in the essential oils of various plants, including bay, cannabis, and hops [12,13]. Monoterpenes are dimers of isoprenoid precursors. It receives its name from the plant myrcia from which it is primarily semi-synthetically produced. It serves as a crucial intermediary in the creation of a number of scents. Myrcene was initially polymerized using Ziegler-Natta-type catalysts in 1960, but sadly the 1,4-stereoregularity was not established [14]. Prior to now, the synthetic processes for polymyrcene included controlled radical polymerization [15,16,17], anionic polymerization with green solvents [18], and cationic polymerization with triflate esters as an in-situ created starting system [19]. Chain transfer causes free radical polymerization to fail, producing branched/cross-linked products [20,21,22,23]. The materials with the highest conversions (almost 100%) were those that had 85% or more 1,4-cis enchainments and 15% or fewer3,4-defects, according to an anionic initiator [24].

The last few decades have seen a rise in interest in nanocomposites, a new class of materials strengthened by nanoscale particles. Toyota researchers first became interested in these novel materials in the early 1990s. In fact, they demonstrated a notable improvement in dimensional stability by dispersing clays in polyamide-6 through in situ polymerizations [25]. In several scientific disciplines, these findings have opened up new possibilities for polymer matrix nanocomposites [26]. Others built on eco-friendly ingredients have replaced toxic polymer-based nanocomposites in recent years. The physicochemical characteristics of the resulting nanocomposites are improved when a polymer matrix is used and a specific quantity of clay is added as reinforcement [27,28]. Two distinct types of nanocomposites structures, intercalated and exfoliated nanocomposites, can be created depending on the degree of interaction between the modified polymer and the clay. Nanocomposites can be created using a variety of techniques, including in-situ polymerization, solution blending of polymers, and others [29].

It is highly intriguing to employ ultrasound to create nanocomposites based on polymer and clay solutions. With the majority of them focused on exfoliating the packed layers of clay, this method of synthesis was utilized to speed up the reaction time and increase the dispersion of nano-fillers in the polymer matrix [30,31,32,33]. The use of ultrasound also affects the morphology of the prepared nanocomposites, particularly in dispersion, such asin-situ polymerization methods [34]. Ultrasonication induces acoustic stream and cavitation bubbles that then undergo an implosion process, releasing heat and energy as a result. This results in a highly well-dispersed reaction medium [35]. For these reasons, many clay and polymer-based nanocomposites were created using an ultrasound-assisted technique.

The use of AlCl_3_ as a catalyst in the synthesis of copolymers based on myrcene and limonene is essentially nonexistent, according to our review of the literature. The main goal of this research is to examine the catalytic properties of AlCl_3_ as a novel catalyst for the copolymerization of myrcene and limonene and Mag-CTA^+^ as a novel, nano-reinforcing filler for the synthesis of nanocomposites based on myrcene-limonene copolymer using ultrasound-assisted method. In our previously published work, we have demonstrated the benefits of various applications of this type of nano-reinforcing filler in several syntheses of nanocomposites in order to improve both the thermal and mechanical properties of the obtained copolymer [36,37,38,39,40].

## 2. Materials and Method

### 2.1. Materials

In this work we have used Myrcene (97%), (R)-(+)-Limonene (97%), AlCl_3_, Methanol (CH_3_OH, 99.9%), Dichloromethane (CH_2_Cl_2_, 99.8%), Sulfuric acid (H_2_SO_4_), Sodium chloride (NaCl), and cetyltrimethylammonuim bromide (CTAB) were purchased by Sigma Aldrich and used as received. Maghnite (Algerian montmorillonite) is supplied in the raw state by ENOF Bental Spa of the National Company of Nonferrous Mining Products, Maghnia Unit (Algeria). The ultrasonic apparatus used to create Mag-CTA+ and nanocomposites is made up of a jacketed glass tank with an ultrasonic horn (13.6 mm in diameter, with a non-replaceable titanium alloy Ti-6Al-4V tip) and a Sonics VC-750 Vibra 6 Cell generator.

### 2.2. Preparation of Mag-Na^+^ and Mag-CTA^+^

Mag-Na^+^ was prepared according to the protocol described by Derdar et al. [41,42]. The raw-Mag supplied by Bental Spa, is crushed and finely sieved, the activation of Maghnite by sodium was carried out using 1 L of NaCl solution (1M) and 20 g of raw-Mag (2% by weight), the latter is mixed for 24 h at room temperature, finally, the Maghnite was washed several times with distilled water.

The Mag-CTA^+^ activation was carried out using ultrasound for 1 h [43]. Firstly 10 g of Mag-Na^+^ is placed in a 1 L Erlenmeyer flask with the chosen concentration (1 CEC). At the end of the exchange process, the suspension is filtered and then washed several times with distilled water. Finally, the solid obtained is dried at 105 °C for 24 h and ground (Figure 1). The structure of organophyllic clay is confirmed by Fourier Transform Infra-red Spectroscopy (FT-IR) and X-ray Diffraction (XRD) analysis and their morphological properties are studied by Scanning Electron Microscopy (SEM) and Transmission Electron Microscopy (TEM) analysis.

### 2.3. Synthesis of Copolymers and Nanocomposites

The copolymerization of myrcene with limonene was investigatedand performed at room temperature for three hours in solution using AlCl_3_ as a catalyst, 0.02 mol of myrcene and limonene, are kept under stirring in 10 mL of Toluene, with 5% by weight of AlCl_3_ (Figure 2). The solution changed color to orange when AlCl_3_ was added to start the copolymerization. Then, as the polymerization process continued, it became darker and thicker. AlCl_3_ was removed from the reaction mixture by stirring with 10 mL of HCL (0.1 M) until the orange color disappeared and the organic phase was washed several times with NaOH (0.1 M) and with distilled water. The organic phase was then dried with magnesium sulphate (MgSO_4_) and toluene was evaporated. The products were dissolved in THF, precipitated in cold methanol (MeOH), and dried in a vacuum overnight. Table 1 summarizes the operating conditions of the copolymerization.

Myr-co-lim/Mag nanocomposites were prepared using a polymer and clay combination synthesis technique in solution. In 15 mL of dichloromethane (CH_2_Cl_2_), 0.5 g of the obtained copolymer is dissolved. The copolymer is then fully dissolved by stirring the mixture for an additional 15 min. The solution is then treated using an ultrasonic-assisted technique for 3 h after receiving 1% by weight of Mag-CTA^+^ [43]. The resulting nanocomposite is filtered, precipitated in MeOH, and overnight vacuum-dried. The same process is carried out again for Mag-CTA^+^ additions of 4, 7, and 10% by weight to myr-co-lim. The samples were given the names myr-co-lim/Mag (see experimental conditions in Table 1).

### 2.4. Characterization

The structure of the obtained copolymer is confirmed by Proton Nuclear Magnetic Resonance Spectroscopy (^1^H-NMR) using a Brucker-Avance 400 MHZ apparatus with Deuterated Chloroform as the solvent. The functional groups of the obtained copolymer, the modified clay, and the obtained nanocomposites were analyzed by infrared spectroscopy using a BRUKER ALPHA Diamond-ATR spectrometer. Differential scanning Calorimetry (DSC) was used, to study the thermal properties of myr-co-lim copolymer, using a204 F1, NETZSCH equipment, operating at a heating rate of 20 °C/min, from room temperature up to 600 °C under an inert atmosphere with a flow rate of 50 mL/min. The morphology of the modified clay and the prepared nanocomposites were observed by XRD diffraction patterns using a Bruker AXS D8 diffractometer (Cu-K radiation), and FEG-SEM on a, JEOL 7001F electron microscopy together with Transmission electron micrographs obtained using a Hitachi 8100. Thermogravimetric analysis (TGA) using a PerkinElmer STA 6000 under nitrogen in the temperature range of 30–700 °C with a heating rate of 20 °C/min, was used to study the thermal properties of the obtained nanocomposites.

## 3. Results and Discussion

### 3.1. Characterization of Clay (Mag-Na^+^ and Mag-CTA^+^)

Figure 1 displays the FT-IR spectra of Mag-Na^+^ and Mag-CTA^+^. We notice two bands at 455 and 515 cm^−1^ that are related to the elongation vibrations of the Si-O-Si and Si-O-Al bonds, respectively, as well as a strong peak at 1057 cm^−1^ [44,45]. The Maghnite’s Si-O vibration is what causes the band to vibrate at 1000 cm^−1^. Following the alteration of Maghnite by CTA^+^, various bands were produced. For example, two new bands were seen for Mag-CTA^+^ in the 2850 and 2922 cm^−1^ regions, which correspond to the C-H stretching vibrations of the methyl group. The intercalation of the alkyl ammonium ions of the CTAB between the clay sheets is confirmed by the FT-IR study results.

The X-Ray diffractograms of Raw-Mag, Mag-Na^+^ and Mag-CTA^+^ are shown in Figure 2. Basal spacing (d001) was determined from XRD patterns using the Bragg equation (2dsinθ = nλ.) where θ is half the scattering angle and λ is the incident beam wavelength; it is 1.01 nm for Raw-Mag. Mag-Na^+^ and Mag-CTA^+^ diffractograms demonstrate that the basal spacing (d001) varies from d = 1.23 nm for Mag-Na^+^ to d = 1.8 nm for Mag-CTA^+^. This growth supports the CTAB’s alkyl ammonium ions’ intercalation in the inter-foliar galleries. These outcomes demonstrate how Mag-CTA^+^ was affected by ultrasonic irradiation.

In contrast to Aicha Khenif et al. [46], who achieved an interlayer distance of 1.98 nm after 24 h of stirring to prepare CTAB/Clay, we were only able to achieve an interlayer distance of 1.8 nm after 1 h.

### 3.2. Characterization of the Obtained Copolymer (lim-co-sty)

#### 3.2.1. H- NMR Measurements

Figure 3 displays the ^1^H-NMR spectrum of the produced copolymer. The proposed structure was verified and further investigated using the ^1^H-NMR spectra. A signal at 0.8 ppm as well as multiple peaks corresponding to the protons of the methyl group is clearly visible in the ^1^H-NMR spectrum of (myr-co-lim). The resulting copolymer’s spectrum also shows the peak (e) at 1.16 ppm, which corresponds to the protons of the methylene group (-CH2); limonene’s spectrum does not show this peak (see Figure 4). The characteristic resonance of the protons caused by the internal double bonds (-CH=CH-) of limonene and myrcene is represented by peaks (b, c and d) between 4.5 and 5.4 ppm which clearly shows that the copolymerization of myrcene with limonene is successful using AlCl_3_ as a catalyst.

#### 3.2.2. FT-IR Measurements

FT-IR measurements were also used to establish the copolymer’s structure. Figure 5 shows the FT-IR spectra of limonene (a) and the produced copolymer (b). The peaks in the copolymer’s spectrum that correspond to the double bonds in limonene at 1309, 1217, 956, 913, and 885 cm^−1^ have disappeared, indicating that the copolymerization process was successful. It should be noted the band corresponding to the stretching band of C=C at 1640 cm^−1^ in the spectra of limonene has decreased in the spectra of myr-co-lim and also an intense band at 2930 cm^−1^ corresponding to the valence vibration of the methylene C-H. A band at 886.38 cm^−1^ corresponds to the valence vibration C-H bond of CH_2_ out-of-plane. The bands at 1456 and 1365 cm^−1^ are attributed to the deformation of the C-H bond of the CH_2_ and CH_3_ groups [47]. These results confirm the copolymerization of myrcene with limonene and also confirm results obtained by NMR analysis.

#### 3.2.3. Differential Scanning Calorimetry (DSC)

The thermal characteristics of the produced copolymer were investigated using differential scanning calorimetry (DSC). Figure 6 shows the myr-co-lim DSC curve. The copolymer’s glass transition temperature (Tg), as determined by its DSC curve, is found to be between 40 and 45 °C. Additionally, the successful copolymerization of myrcene with limonene usingAlCl_3_ as catalyst is demonstrated by comparing the Tg of the copolymer with those of polmyrcene (Tg) in the temperature range of −73.09 to −65.90 °C [48] and polylimonene (Tg = 116 °C) [49].

### 3.3. Characterization of the Obtained Nanocomposites (myr-co-lim/Mag)

The XRD patterns of the Mag-CTA^+^ copolymer and nanocomposites are shown in Figure 7. We noticed that the copolymer we obtained (myr-co-lim) does not exhibit any distinct peaks in its XRD pattern, indicatingan amorphous structure. The characteristic basal diffraction peak of Mag-CTA^+^ at 2θ = 4.9° was almost completely absent in the case of myr-co-lim/Mag 1%, demonstrating the exfoliation of the clay, which accounts for a good diffusion of myr-co-lim copolymer in the clay galleries. In this case, the nanosheets are all individualized and occupy the entire volume of material. The nanocomposites made by adding 4, 7 and 10% of Mag-CTA^+^ had a single which corresponded to interlayer distances of 2.15, 2.62 and 3.27 nm. In comparison to the Mag-CTA^+^, which had an interlayer distance of 1.8 nm, these nanocomposites’ interlayer distances were enhanced by a factor of more than two. The increase ininterlayer distance indicates that there is an intercalation of the copolymer into the Mag-CTA^+^ sheets and an agglomeration of leaves intercalated through interactions between the sheets of Montmorillonite. This result confirms that the copolymer was well intercalated between the clay galleries. These results are in agreement with those obtained by Hanène Salmi-Mani et al. [50].

Figure 8 displays the produced nanocomposites’ FT-IR spectra (myr-co-lim/Mag 1, 4, 7, and 10%). We noticed that the produced nanocomposites exhibit vibration bands that almost exactly match those of the organo-modified clay (Mag-CTA^+^) and are in good agreement with the pure copolymer structure. The FT-IR spectra of the produced nanocomposites revealed the double bond C-H in the copolymer at 2921 and 2867 cm^−1^, respectively which corresponds to the vibration of the methyl and methylene group. The FT-IR spectra of the produced nanocomposites display the appearance of a strong peak at 1000 cm^−1^ that corresponds to the vibration of Si-O of the Mag-CTA^+^ in contrast to the FT-IR spectrum of the pure copolymer, theseresults are due to the different amounts of clay used in the preparation of nanocomposites. These findings demonstrate the intercalation of myr-co-lim copolymer between the clay sheets.

Figure 9 displays the SEM images of the produced nanocomposites (myr-co-lim/Mag 1, 7, and 10%) as well as the Mag-CTA^+^. We noticed a more structured montmorillonite structure in small particles when comparing the morphology of Mag-CTA^+^ with myr-co-lim/Mag 7 and 10% nanocomposites. The examination of nanocomposites at 10 m showed the creation of separated montmorillonite plates, which is a partial exfoliation, as well as a rougher surface and a covering of the montmorillonite surface by the copolymer in the myr-co-lim/Mag 1% nanocomposites.

Figure 10 displays the images acquired using TEM of Mag-CTA^+^ and the nanocomposites that were produced. The dispersion of Mag-CTA^+^ in the copolymer matrix was ascertained by TEM analysis, which was also used to validate the outcomes of the XRD investigation. The dark and brilliant lines make it simple for Mag-CTA^+^ to identify the silicate layers. The clay nanoparticles are mostly evenly spread in the copolymer matrix of the nanocomposites made with 1% by weight of Mag-CTA^+^, and they exhibit a partially or completely exfoliated structure. However, the nanocomposites of myr-co-lim/Mag 7 and 10% exhibit a modified clay intercalated structure. These findings support those of the XRD analysis. These results are in agreement with those obtained by H Derdar et al. [51].

Figure 11 displays the TGA curves of the produced nanocomposites and pure copolymer. We found that pure copolymer and all nanocomposites have a one-step process for weight loss. The thermal stability of the produced nanocomposites is improved by the Mag-CTA^+^, according to TGA curves. While the degradation temperature of the pure copolymer was observed at 120 °C, it can be seen that nanocomposites prepared with 1 and 10% by wt of Mag-CTA^+^ exhibit high thermal stability up to a degradation temperature of about 200 °C. The richer the nanocomposite is in copolymer, the quicker it is degraded. According to previous research [52], the creation of a protective carbonized layer is what causes this increase in thermal stability. The fine dispersion of intercalated or exfoliated clay particles, which serve as an inorganic support, aids in the creation of this layer [53]. In general, the insertion of exfoliated lamellar silicates raises the degradation temperature of the polymers, increasing their value and enabling their use at higher temperatures [54,55,56].

## 4. Conclusions

With the use of AlCl_3_ as a catalyst, myrcene and limonene were successfully copolymerized and the copolymerization proceeds by a cationic mechanism and the results are outstanding. The structure of the resulting copolymer is confirmed by H^1^-NMR, FTIR, and DSC analysis. It also investigated how the characteristics of myr-co-lim/Mag nanocomposites are affected by the preparation and application of organomodified clay (Mag-CTA^+^) in various ratios. The research demonstrates that the varied organoclay (Mag-CTA^+^) ratios have an effect on how nanocomposites copolymer/clay are formed. The nanocomposites prepared with 1% by weight of Mag-CTA^+^ were exfoliated, according to the FT-IR and XRD data, while those prepared with 4, 7, and 10% wt of Mag-CTA^+^ were intercalated, expanding the interlayer gap between the layers. SEM and TEM studies confirmed an organization of some particles and in other cases, a separation in plates made up of montmorillonite layers, this supports partial or complete exfoliation of montmorillonite in the copolymer matrix and development of the nanocomposites. According to thermogravimetric data, the nanocomposites exhibit greater thermal stability than pure copolymer (T 200 °C). The main goal of this research is to create copolymers and nanocomposites using terpenes and clay as green raw materials. The obtained nanocomposites (myr-co-lim/Mag) will use in different areas such as cosmetics, medication, and even as a pollution absorber.

## Data Availability

Not applicable.

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
