# Peer review of "Green Copolymers and Nanocomposites from Myrcene and Limonene Using Algerian Nano-Clay as Nano-Reinforcing Filler"

_polymers, 2022, doi:10.3390/polym14235271_

Round 1

Reviewer 1 Report

Derdar et al. report on the “Green Copolymers and Nanocomposites from Myrcene and Limonene Using Algerian Nano-Clay as Nano-reinforcing Filler” The content of the work is interesting, but the manuscript cannot be published in the present form due to the following issues:

1.      Reaction steps are absent for the preparation of in the section “Preparation of Mag-Na+ and Mag-CTA+

2.      In the XRD figure 2 the planes should be highlighted.

3.      Author should explain in detail the XRD peak shift in Figure 7, its cause and the reasons for the change in the FWHM.

4.      EDS is absent from the SEM images.

5.      Explanation of the FTIR results is compulsory. There is no explanation of the modifications in the sharp peak intensity from 1200-1350 cm-1

6.      SAED pattern is absent for the confirmation of the product which is highlighted in Figure 10 for Mag-CTA+ (a), myr-co-lim/Mag 1% (b), myr-co-lim /Mag 7% (c), myr-co-lim /Mag 10%.

7.      Font size used in Figures 2, 6, 7 and 11

Author Response

Our Responses to the Reviewer Comments

We thank all three reviewers for their careful review of the manuscript and their suggestions for enhancing it. We have responded positively to all of the suggestions where possible

Reviewer 1

  1. Reaction steps are absent for the preparation of in the section “Preparation of Mag-Na+ and Mag-CTA+”

Reaction steps for the preparation of Maghnite (Na and CTA) are represented in Schem 1 in the revised paper.

  1. In the XRD figure 2 the planes should be highlighted.

Figure 2 has been modified in the revised paper.

  1. Author should explain in detail the XRD peak shift in Figure 7, its cause and the reasons for the change in the FWHM.

The Discussion is more detailed in the revised paper (corrections are in highlighted in yellow)

  1. EDS is absent from the SEM images.

We agree with your comment, but the instrument  we used did not have EDS option.

  1. Explanation of the FTIR results is compulsory. There is no explanation of the modifications in the sharp peak intensity from 1200-1350 cm-1

See the revised paper.

  1. SAED pattern is absent for the confirmation of the product which is highlighted in Figure 10 for Mag-CTA+ (a), myr-co-lim/Mag 1% (b), myr-co-lim /Mag 7% (c), myr-co-lim /Mag 10%

I totally agree with you, however due to the short time available for the revision of a manuscript Polymers journal we are not able to add the SAED analysis.

  1. Font size used in Figures 2, 6, 7 and 11

Figures are simply modified in the revised paper.

Reviewer 2 Report

In this report, the authors introduced a novel method to synthesize myrcene-limonene copolymers and nanocomposites with green materials. The intermediates and products are characterized by standard lab instruments and presented clearly in this manuscript. Overall this is a good paper to publish on Polymers.

Typo: line 135: 0,02 mol -> 0.02 mol

Author Response

Reviewer 2

We note the very positive comments made by this reviewer.

  1. Typo: line 135: 0,02 mol -> 0.02 mol

Corrected

Reviewer 3 Report

Dear Authors,

My comments was attached as a file. 

Author Response

Reviewer  3:

  • The chemical and physical properties of the clay used should be detailed in the material section. The nanoparticle structure should be specified.

The proposed structure of the clay used in this work is represented in the revised paper in Scheme 1.

  • plagiarism ratio is really very high, It should be definitly revised aproximately %50 increadible.

We have reduced the  level of plagiarism in the revised paper.

  • Extraction of limonene could have been more impressive in this study, for example from citrus fruits. We can see that some literatures use AlCl3 and FeCl2 catalysts in copolymer synthesis. In this study, the amount of AlCl3 used as a catalyst was preferred as 5%. For this, any optimization was made before the study and why 5% was used should be explained.

First, we have several works using limonene, we’ll try to use an extracted limonene in the next time.

We used 5% of AlCl3 regarding to our previous work about the polymerization and copolymerization of limonene using clay as catalyst and by examing the literature we found that the use of lewis acids as catalyst should not be more than 5% by weight.

  • In the X-Ray analysis results, 1.8 nm interlayer distance was formed with 1 hour of ultrasonic treatment, and this is a really good score. I don't understand why this time was not used longer.

We prepared the organophilic clay using the protocol used by Cherifi Z et al and the work was cited in the revised paper, reference [43]. The use of ultrasonic assisted method is great advantage but if we use it for long time we could destroy the clay structure.

  • Part of 3.2.2. FTIR measurement (214-224) need to support with literatüre I can recommend the following article. Investigation of antimicrobial properties of QASs+ (novel synthesis) G Baysal, H Aydın, S Uzan, H HoÅŸgören Russian Journal of Physical Chemistry B 12 (4), 695-700.

Thank you. We add this reference in the revised paper.

  • The relationship between the Tg value and the catalyst activity used in the DSC analysis results should be discussed by adding the literature.

Tg of the obtained copolymer was confirmed regarding to previous work (see references 48 and 49 in the revised paper).

  • As it is known, exfoliated structures of nanocomposites are more preferred. The analysis results obtained show an exfoliated structure except for 1% clay additive. The advantage of this and the spectra in the field of use should be specified.

The exfoliated structure is obtained by using a low amount of clay see reference 50 in the revised paper, in our work we used 1% of clay and we obtained an exfoliated structure. XRD, SEM and TEM analysis confirm the exfoliated structure. 

  • The nanocomposite structure doped with 1% clay formed exfoliated structures, but intercalated structures occurred for 4.7% and 10%. I'm wondering if it's aggregation. These divergent results need literature support.

In XRD analysis of the obtained nanocomposites we support the results with the work by Hanne Salmi-Mani et al (reference 50) and our obtained results are in agreement with those obtained by Hanène Salmi-Mani et al.

  • The usage areas of the final products and the advantages they will provide should be written in the conclusion section.

The usage areas of the final product are provided in the revised paper.

  • the following statement needs literatüre; “The clay nanoparticles are mostly evenly spread in the copolymer matrix 278 of the nanocomposites made with 1 % by weight of Mag-CTA+ , and they exhibit a partially 279 or completely exfoliated structure. However, the nanocomposites of myr-co-lim/Mag 7 280 and 10% exhibit a modified clay intercalated structure. These findings support those of 281 the XRD analysis”

We have added a reference in the revised paper.

Round 2

Reviewer 1 Report

The revised manuscript is acceptable for publication

Reviewer 3 Report

Dear Authors,

I am pleased to accept  your revised manuscript